# Semantic Rheology: The Flow of Ideas in Language Models

## Abstract

The flow of ideas has been extensively studied by physicists, psychologists, and machine learning engineers. This paper adopts certain tools from microrheology to investigate the similarity-based flow of ideas. We introduce a random walker in the word embeddings and study its behaviour. Such similarity mediated random walks through the embedding space shows signatures of anomalous diffusion, commonly observed in complex structured systems such as biological cells and complex fluids. The paper concludes by proposing the application of popular tools employed in the study of random walks and diffusion of particles under Brownian motion to quantitatively assess the incorporation of diverse ideas in a document. Overall, this paper presents a self-referenced method that combines concepts from microrheology and machine learning to explore the meandering tendencies of language models and their potential association with creativity.

## 1 Introduction

The random walker was introduced to the world by Karl Pearson in 1905. Pearson posed the question of determining the probability of a person starting from a point and taking a series of straight-line walks, and then calculating the likelihood of ending up within a certain distance from the starting point. Random walks have since garnered significant attention due to their association with Brownian motion and diffusion processes, making them a well-studied subject across numerous disciplines.

### 1.1 Random walk through similarity space

In this paper, we introduce a random walker in the word embedding space. Our objective is to study the flow of ideas lead by semantic similarity. The flow of ideas has been closely linked to quantifying creativity. Prior research has already highlighted the significance of correlating and processing diverse topics as an indicator of creativity. For instance, studies by Olson et al. (2021) have demonstrated that the ability to associate unrelated words can predict creative thinking. Moreover, semantic distance has been utilized as a measure of creativity in human studies in the field of psychology (see: Kenett (2019), Beaty & Kenett (2023),Heinen & Johnson (2018)).

### 1.2 Shakespeare and HG Wells

To quantitatively assess the ability of a document to interconnect diverse ideas, we draw upon a popular tool employed in the study of random walks and diffusion of particles under Brownian motion. Such analysis may offer valuable insights into the dynamics and patterns of idea flow, providing a means to explore and analyze the creative tendencies of language models. In case of sentence embeddings for a document, a mean squared distance curve can be defined for different intervals between the position of the sentences in the text corpus. In Figure 1, this is represented as the delay in the x-axis.

Assume a document containing $N$ sentences. The $k^{th}$ sentence can be embedded to a vector represented by $\vec{e_k}$. The distance between two sentence pairs can be measured as:

$$d_{ij} = \vec{e_i} - \vec{e_j}, 1 \leq i < j \leq N$$

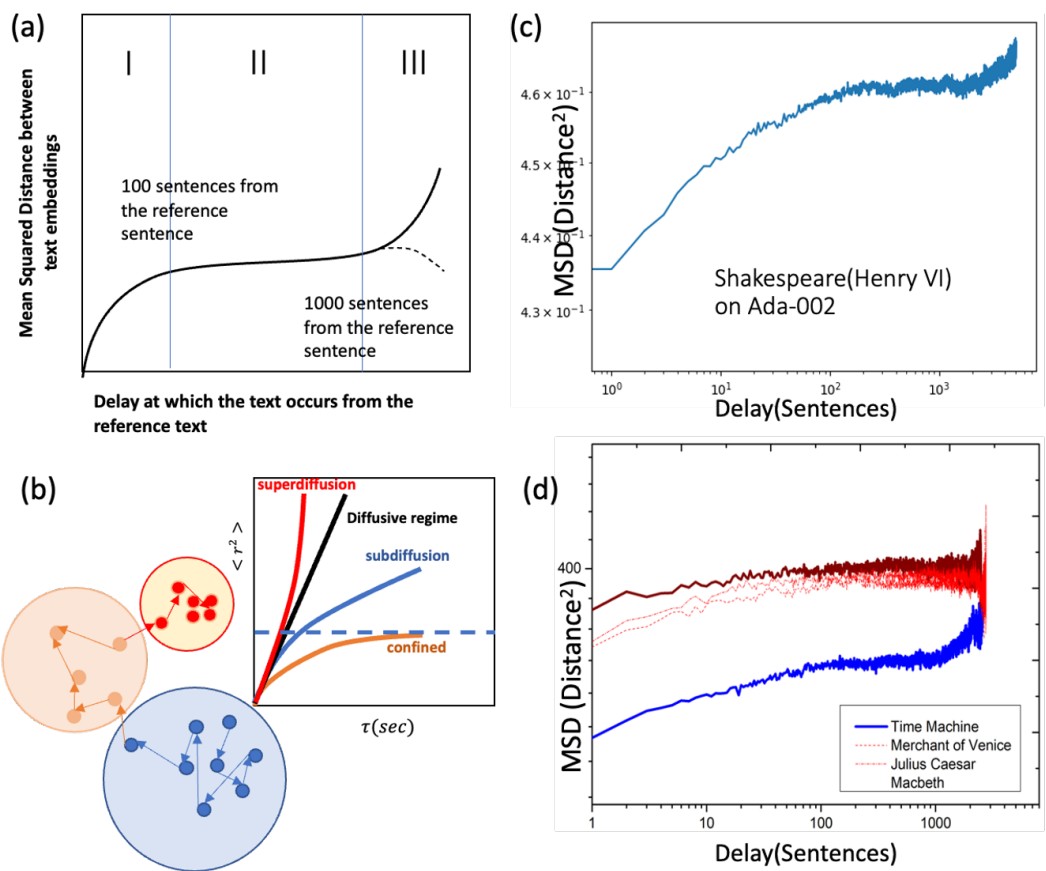

Figure 1: (a) A representative shape of the typical MSD curve observed for large language corpus. (b) Inset graph shows general behavior of MSD curves for diffusing particles. These behaviours has been adapted for this paper to explain behaviour of similarity walks. The particle trajectory represents brownian motion of particles in confined strucures with narrow probability of escape.(c) MSD of a Henry VI text corpus using OpenAI's text-ada-002 embeddings. Some equivalence with the hypothesized behaivour in (a) is observed. (d) MSD curve of popular writings by Shakespeare and H.G. Wells demonstrating similar behavior

where $i$ and $j$ are the location of the sentences in the document. The delay is simply:

$$\Delta(sentence) = j - i$$

There are thus many distinct distances for small delays and very few for large delays. For a purely random language meandering through a sufficiently large corpus of text, where the sentences or words appear without any correlation to previous sentences, the mean displacement would be analogous to diffusion due to Brownian motion. The interesting statistical quantity to consider here is the mean squared distance (MSD) of the sentences at given delays. Mean squared displacement analysis is often used to study the type of diffusion regime undergone by a diffusing particleMichalet (2010). The MSD for a particular $n$ sentence delay for the corpus with $N$ sentences can be defined as:

$$MSD(\Delta_n) = \frac{1}{N-n} \sum_{i=1}^{N-n} (\vec{e_{i+n}} - \vec{e_i})^2, n = 1, ..., N-1$$

With this definition, it is easy to observe some patterns in MSDs of well-known novels. We hypothesize that similar patterns can be found in most creative endeavors in human literature. As shown in Figure 1(a) , in short delays between sentences the mean squared distance shows a ballistic rise. This

phase has been marked as phase I in the MSD graph. In this phase, about 10 to 100s of subsequent sentences are used to explain the reference sentence. Here, new ideas, characters and story plots are quickly introduced. In attempting to provide context and new information about the reference sentence, the author tends to use words and sentences that are somewhat semantically distant from the reference sentence.

The phase marked as II in Figure 1(a), can be considered more as a consolidation phase. This is the part where the main story plays out. The same characters and plots are discussed in more details in 100 to 1000 sentences after their introduction. Phase III can be considered as the surprise or creative phase. At a delay of a few thousand sentences, the climax or the surprise occurs for the reference sentence. We have occasionally observed a return to the initial semantic distance which can be attributed to ending passages where the initial characters or plot are revisited after the story is complete.

This can be contrasted with random walks of particles as shown in Fig 1(b)

We extend this analysis to see if similar patterns hold for primitive word embeddings and generative language models.

## 2 PRETRAINED WORD2VEC

Word2Vec Mikolov et al. (2013)is a neural network model that encodes semantic information of the words in the corpus given an unlabeled training data. Semantic similarity between word vectors is calculated by their cosine similarity. We use this model due to the relative ease of using pretrained models available in gensim library.

### 2.1 FREE FLOW FROM SINGLE WORDS AND 'JABBERWOCKY'

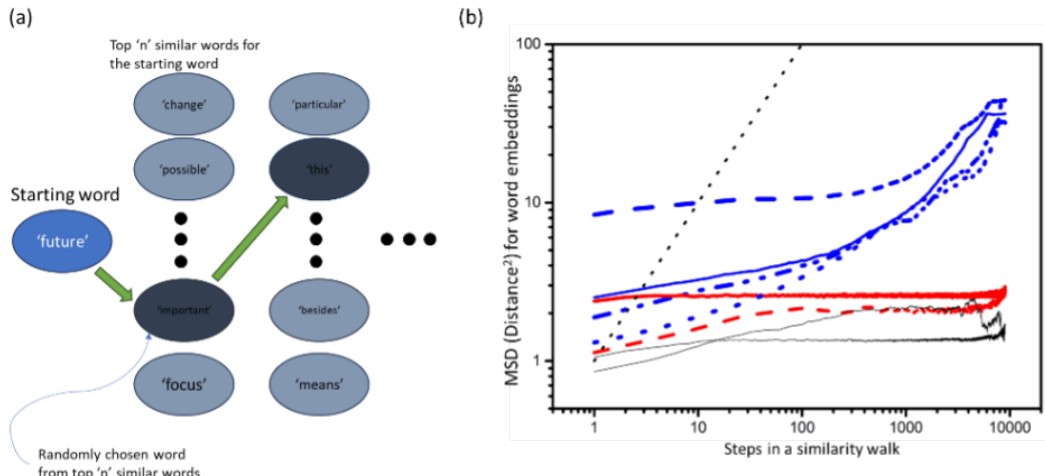

Figure 2: (a) Representation of a similarity walk. In a similarity walk, top 'n' similar words are searched for, following which a word is randomly chosen and the same is repeated for 'k' steps to complete a 'k' step similarity walk. (b) MSD curve of a few similarity walks undertaken in pretrained Word2Vec models.

To test if language models have similar properties as described in 1.2, we try a 'similarity walk' through a pretrained word embedding model. We use Word2Vec, a simple language model that positions word embeddings in a vector space, where semantically and syntactically similar words are located close to each other, while dissimilar words are farther apart (source: ).

A 'Similarity walk' can be defined as a process where a word is randomly chosen or provided as an input by the experimenter. From the pretrained model, a list of top 'n' most similar words is identified, and a word is randomly chosen from those 'n'-similar words for the next step. If this

process is repeated 'k' times, it would be considered a k-step similarity walk through the corpus. This is schematically represented in Figure 2(a). The objective is to identify patterns in how a one-word idea might flow within a language model.

We observe that by following similar words, word vectors eventually converge into a cluster of closely linked obscure words, leading to a path from which the probability of finding newer words through a similarity walk becomes almost zero.

In Figure 2(b) we do a MSD analysis of the words encountered in the similarity walk which uphold the above observation. The starting words chosen purposefully by the author are colored blue and represent a biased choice for the starting idea. The MSD curve in red and black were randomly selected from the corpus, and are more representative of the type of behaviour a similarity walk exhibits. The difference between biased choice and random choice for starting words in highlighted because it was often observed that common english words often discovered more unique words during a similarity walk compared to a randomly chosen obscure word. This observation is empirical and does not hold true in general.

The interesting feature of these MSD plots are the near flat curves or the eventual approach to flatness over long distance for the blue curves. This behavior is observed for sufficiently long similarity walks, irrespective of the starting word. The walk always converges into a cluster of words that are unrelated to the starting words but are closely connected by similarity to each other. After a similarity walk falls into these clusters of words, it is unlikely to discover any new words by continuing the walk. The path to this obscure cluster of topics appears to follow a Levy flight pattern, where word vectors oscillate around similar ideas and occasionally take long-distance leaps in semantic space. This descent into a 'jabberwocky' of words is an universal property of all language models we have tested so far. This limitation appears due to the embedding of a few words that are rarely used anywhere else in the corpus. For example in the fasttext-wiki-news-subwords-300 corpus, most of the walks would fall into a cluster of usernames and mail addresses that refer to each other A representation of this behavior in dimensionally reduced projections for fifty different starting words over three different pretrained corpus is shown in Figure 3.

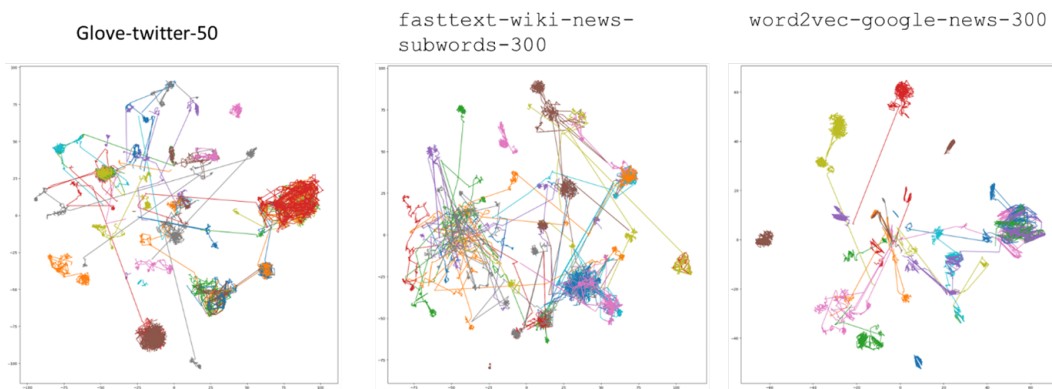

Figure 3: tSNE visualization of fifty similarity walks on pretrained Word2Vec models. The models include: "Glove-twitter-50', 'fasttext-wiki-news-subwords-300' and 'word2vec-google-news-300'. The walk structure in 2D feature space appears to mimick foraging behaviour where the forager occasionally takes a long Levy flight.

## 2.2 GUIDED SIMILARITY WALK

The results in 2.1 represent an unrestricted motion through the embedding space with single words. However, modern language models rarely work with single words. A better understanding of the dynamics of similarity walk can be found with guided similarity walks. To simulate guided walks in Word2Vec models we use multiple words as tethers to find the next similar word.

As shown in Figure 4(a) the most similar words for the next step in a guided similarity walk consists of an original idea along with two to three other guiding words. This reduces the total vocabulary

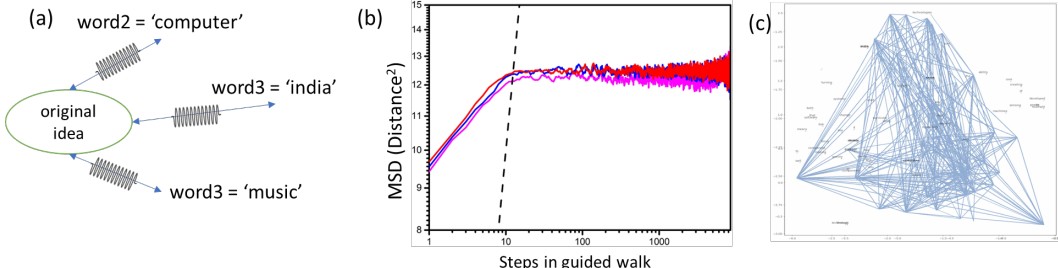

Figure 4: (a) A representation of guided similarity walk, where the next similar words are decided by the starting word along with other words, thereby restriciting the choices of words that can be discovered through a walk. (b) MSD curve of such similarity walks show highly confined behavior. (c) tSNE representation demonstrating how the walk become more closed and the lack of foraging behavior.

available for a similarity walk. The constraints placed on similarity walk by the guiding words behavior shows up in the MSD plot as a confined motion as shown in Figure 4(b). It is also interesting to observe that the magnitude of the distance moved during this walk is comparatively smaller. This indicates a very tight confinement in the embedding space, even though we experimented with words that are supposed to be semantically distant. The TSNE plot in Figure 4(c) is a low-dimensional representation of how the nature of the random walk has changed under constraints. While a freely diffusing word showed Levy Flight like behavior, a constrained walk looks more like a closed loop.

## 3 SIMILARITY WALKS WITH SENTENCE TRANSFORMERS

Natural language processing has seen a paradigm shift with the introduction of the transformer model. The astounding progress in generating coherent replies to user queries have been largely due to the dense and rich sentence embeddings. In this section of the paper we look at similarity walks through sentence embeddings.

### 3.1 SIMILARITY WALKS THROUGH SENTENCE TRANSFORMERS

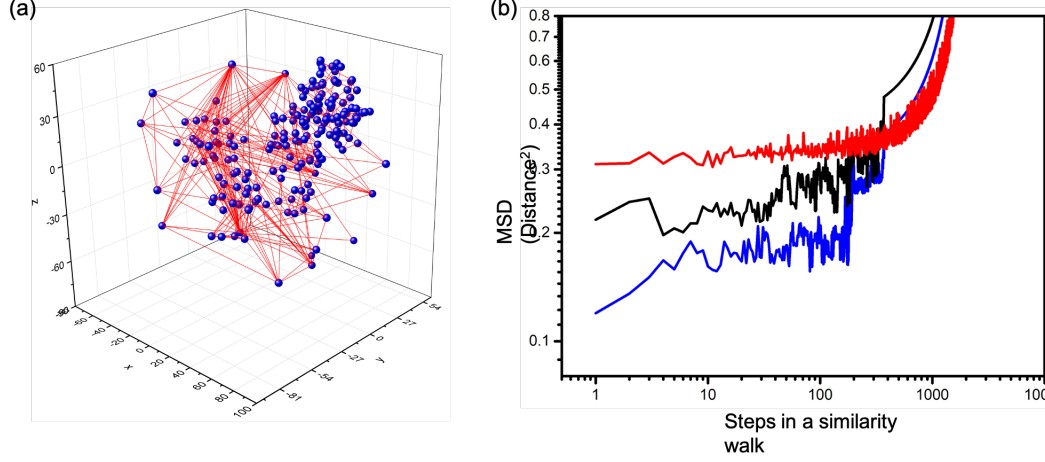

Figure 5: (a) A tSNE representation of similarity walk through sentence transfomers (b) MSD curve showing a confined behavior for approximately 300 sentences, beyond which the similarity walk falls into a jabberwocky.

To perform a similarity walk thorough sentence embeddings we used the distilBERT model Sanh et al. (2019). The corpus for our experiment was chosen from among the books in Project Gutenberg throught the NLTK library Bird et al. (2009) Sentence Transformers show similar behavior where after about 300 sentences, the recursive querying for similar sentences in the corpus eventually converge into a cluster of closely linked sentences that carry very little semantics and are obscure in the context of the corpus. An example of this behavior can be found in the supplementary csv files. In Figure 5(a), a dimensionally reduced visualization of this behaviour can be seen. Similarly, divergence from the coralled behaviour in the MSD as seen in Figure 5(b) also highlights this pattern.

Similar experiments were also performed on large language models like alpaca-lora-7b. The results are available at Appendix A. With large language models generating similarity walks was computationally challenging and will be explored further in a later paper.

## 4 DISCUSSION

In this paper, we introduce a random walker in the word embedding space to study the flow of ideas lead by semantic similarity. We find that the random walker exhibits signatures of anomalous diffusion, which is commonly observed in complex structured systems such as biological cells and complex fluids. This suggests that the flow of ideas in language models is not random, but rather follows a complex pattern.

We study the MSD curve of the embedding vectors to quantitatively assess the incorporation of diverse ideas in a document. We find that the diffusion of ideas in language models is correlated with the document's creativity. This suggests that the ability of language models to generate creative text is related to their ability to flow ideas in a complex and nuanced manner. This paper presents a self-referenced method that combines concepts from microrheology and machine learning to explore the meandering tendencies of language models and their potential association with creativity. The correlation between the diffusion of ideas and the document's creativity is stronger for generative language models than for primitive word embeddings. This suggests that generative language models are better able to incorporate diverse ideas and to generate creative text. Similarity walks can be extended to any source of information be it music, art or text. The existence of small closely linked clusters from which the chances of escape is narrow is an interesting part of most language models. In future, we plan to understand the information theoretic implications of such valleys. We plan to study the factors that influence the flow of ideas in language models. We believe that by understanding these factors, we can develop techniques for controlling the flow of ideas in order to generate more creative text.

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
