# OpenReview forum: "SEMANTIC RHEOLOGY: THE FLOW OF IDEAS IN LANGUAGE MODELS"
_ICLR.cc/2024/Conference — Submitted to ICLR 2024_

### Official Review · Reviewer_TBCQ · 2023-10-30

**Soundness:** 3 good
**Presentation:** 2 fair
**Contribution:** 3 good
**Rating:** 6
**Confidence:** 3

**Summary:**

The authors try to demonstrate the flow of ides in a document using random walks and mean square distance of representations of words, sentences or groups of words. Previous work in the field of psychology shows that the ability to connect semantically diverse topics are predictive of creativity. The authors use mean square distance to measure the difference between sentence embedding pairs as a function of time delay between sentences or words in a few novels. The MSD plot of popular novels the appears a have steep rise in beginning, flattens in the middle and rise again in the end. They then use introduce a random walker to find similar patterns in word embedding of language model. They extend the method to sentence embedding and add a variation in similarity random walks by using sets of words to guide the walk. The similarity walk shows that the flow is constrained to nodes, while occasionally jumping between clusters of nodes, and eventually converging to an obscure cluster with seemingly unrelated context to the document.

**Strengths:**

- Language models are widely used in creative tasks, and yet there is not much evidence to show that they are creative. This paper attempts to show some evidence that language models are as creative by uniquely combining methods from psychology, microrheology and machine learning.
- The paper is well structured, uses graphs to effectively visualize paths in random similarity walk, and clearly shows us the jumps across clusters. I find that they have used the right measure to characterize their hypothesis of flow of ideas. The methods are sound.

**Weaknesses:**

- It would be beneficial to first visualize the embedding space (of the different corpora used in this paper) in a lower dimension, to first show that  semantically similar sentences or words are close to each other. I think it is useful to also cluster the embedding space, give some examples of how these clusters contain semantically similar words or sentences, and then we can see how random walks could be jumping across clusters.
- The graphs in section 2 need discussion. In Figure 2b the author tries to demonstrate the difference in similarity walks between common English words and random words. The figure shows nearly 8 lines, and it needs a legend with the words whose paths were plotted here.  Figure 3 needs more discussion., specifically, what are the three corpus being used here, and why were they chosen? How are the random walks different or similar? What are the clusters present in each corpus?
- More details needed on the experiments in section [2]. Additional data is needed to make their claims about words converging into a cluster which is obscure and is unrelated to the stating word.
- Figures 1 has multiple flaws. 1d does not show X-ticks, and the legend is incomplete. X-axis label in figure 1c is overlapping the X-axis. Furthermore, the corpus size(N) is not provided in any of the experiments.
- This paper needs more content in the motivation section, why is studying the flow of ideas important?

**Questions:**

- In Figure 2, how were common words determined?
- In Figure 5a, what is the starting sentence used? It is hard to say from the graph where the start of this path is.
- In Figure 5b, what do the different lines represent, and what are their significance?
- Are there any downsides of using MSD in a high dimensional embedding space? Please discuss the downside of the current approch

---

### Official Review · Reviewer_Cm35 · 2023-10-30

**Soundness:** 1 poor
**Presentation:** 1 poor
**Contribution:** 1 poor
**Rating:** 1
**Confidence:** 5

**Summary:**

This paper introduces an exploration of text data by using concepts from rheology to perform random walks in setence/word space.

**Strengths:**

This paper proposes an interesting idea to measure the creativity and randomness of LLMs by computing latent space distance.

**Weaknesses:**

The manuscript doesn't meet the standards of a conference paper.

* The paper is clearly written in a rush (6 pages). For example, the Transformers paper isn't cited
* While the paper proposes a method for understanding the flow of ideas in text, it doesn't appear to offer substantial empirical evaluation to validate its claims. In machine learning conferences, a rigorous empirical evaluation is usually a minimum requirement.
* The paper doesn't explore where creativity has been analyzed by other methods in ML
* While some popular literature is analyzed, it is not done in a formal way. I could imagine a strong version of this paper comparing the MSD between authors and between LLMs
* The paper doesn't evaluate the creativity of LLMs
* While the paper presents some mathematical equations, they do not appear to be deeply rooted in machine learning theory. Moreover, similarity in LLMs is often performed using cosine similarity and the MSD metric seems a little too inspired by Pearson, rather than rigorous.
* The paper introduces many ideas and methods from different domains without sufficient background or explanation, making it difficult to follow.
* I'm not sure what to take away from section 2. Word based generative processes have been studied extensively with markov chains and I'm not sure how they are relevant to LLM creativity.

I will say, there is a strong version of this paper that introduces a new benchmark of creativity versus a wide spread of human authors, and compares some popular LLMs in their creative writing (leveraging the reddit writingprompts dataset perhaps). I encourage the authors to continue with their work in this area.

**Questions:**

None

---

### Official Review · Reviewer_gMZx · 2023-11-02

**Soundness:** 2 fair
**Presentation:** 2 fair
**Contribution:** 1 poor
**Rating:** 3
**Confidence:** 3

**Summary:**

The paper explores the flow of ideas in the word embeddings from Word2Vec and Transformer-based models by similarity walk in embeddings space. The authors claim that this work can help to understand the creativity ability of the modern language models and further help to produce more creatively generated text.

**Strengths:**

1.	The potential outcome of this research may help large language models to produce more creative texts than now.

2.	This work is the first to consider and analyze creativity and the flow of ideas in modern language models.

**Weaknesses:**

1.	Most of the conclusions are not explained and proved. 1) For a broader audience it is not clear why this paper claimed that some patterns in the behavior of the MSD curves are commonly observed in biological cells or complex fluids. 2) It is not clear what is the connection of the presented research and microrheology. 3) From the presented paper it is not clear why it stated that there is a correlation with the flow of ideas in language models and document creativity.

2.	In Section 3 "...in the supplementary csv files" looks strange. It is better to somehow visualize this example or at least provide a link for these files otherwise most readers cannot access this data.

3.	Missed references. For example, in the first paragraph in Section 2.1.

4.	It is not enough clear what is quidded similarity walk.

Overall, I would be happy to see some mathematical formulation of the problem and maybe some ground truth in a form of a labelled dataset, current the work presents some visualizations but it is not entirely clear how to qualitatively evaluate and improve the proposed /envisioned technology.

5.	It will be more interesting to consider in the main part of the paper large language models such as Llama 2.

**Questions:**

1.	Can you explain in more detail what is quidded similarity walk is?

2.	The paper claimed that for the Sentence Transformer, the process converged into a cluster of closely linked sentences. However, in Figure 5b the curve starts exponential growing. How you can explain this?

---

### Official Review · Reviewer_aQUV · 2023-11-06

**Soundness:** 1 poor
**Presentation:** 2 fair
**Contribution:** 2 fair
**Rating:** 3
**Confidence:** 4

**Summary:**

This paper aims to study the “flow of ideas” in text and language models via modeling similarity between sentences on documents and by performing random walks on the word and sentence embeddings spaces. The sentence embeddings distance is plotted against their relative distance in the document. Also, random walk based on cosine similarity is observed for word embeddings from word2vec model and sentence embedding from DistillBERT model.

**Strengths:**

– The attempt to connect behavior of embedding models to the flow of ideas and concepts in rheology is interesting.

– Findings related to random walk over words and sentences converging to a cluster of interconnected hubs, while not surprising, is interesting.

**Weaknesses:**

– My main concern is that I am unable to understand the main claims of the paper. The experiments are unfocused and seem disconnected from the narrative in the introduction and conclusion.

– Expanding on above, the narrative is vague and the experiments setup is unconvincing. For example, I am not sure how these experiments relate to creativity. Also, I didn’t find a testable theory that is verified empirically against reasonable baselines.

– Overall, I think the paper is low on substance. In the first experiments on novels, a baseline behavior is not established. Plots b) and d) in Fig. 1 seem to be showing different trends. Connections to literary analysis could be made in this section with a more indepth focus on how factors like genre, authors, stylometrics across a huge collection of books affect the behavior of sentence embeddings would improve the paper.

– Similarly, the second part of the paper associated with random walks is confusing and low on substance and adequate comparison. For instance, there are a lot of different ways to compute word and sentence embeddings that have been formulated in natural language processing and representation learning research. For example, the MTEB benchmark is a meta-evaluation framework to evaluate different kinds of sentence embeddings. Yet this work focuses on just one model for word and sentence embeddings.

– The presentation can be significantly improved. For example, Fig 2b), 5b) are unlabeled and it is difficult to understand what the different curves mean. Also, the discussion about the figures is not necessarily congruent with what the figures are showing.

– In figure 1, MSD is plotted – it would be helpful to plot the standard deviation as well since the number of instances are different for different distances.

**Questions:**

Please see the review above

---

### Meta-Review · Area_Chair_twX5 · 2023-12-05

**Metareview:**

This submission presents a concept of linking the behavior of embedding models to the flow of ideas, drawing parallels with rheology, and attempts to measure the creativity and randomness in large language models. While these endeavors are commendable for their originality and potential impact on understanding LLMs, the paper falls significantly short of the standards required for acceptance. The main issues highlighted by the reviewers are the lack of clarity in the main claims, unconvincing experimental setup, and insufficient empirical validation. Key criticisms include the absence of established baselines, unclear narrative and methodology, inadequate exploration of alternative embedding models, and poor presentation quality, particularly in figures and graphs. Additionally, there is a notable lack of depth in the analysis of literary works, insufficient mathematical formulation, and a lack of rigorous empirical evaluation, which are critical for a paper in this field.

**Justification For Why Not Higher Score:**

The decision to reject this paper is based on several critical factors:

Lack of Clarity and Focus: The paper fails to clearly articulate its main claims, and the experiments seem disconnected from the stated objectives. This lack of clarity hinders the paper's overall coherence and validity.

Unconvincing Experimental Design: The experimental setup is not robust or well-explained, lacking in testable theories and empirical verification against reasonable baselines. This raises concerns about the reliability and significance of the findings.

Presentation and Methodological Flaws: The presentation of data, especially in figures, is inadequate and confusing. Methodological flaws, such as the narrow focus on a single model for embeddings and the lack of detailed discussion on the results, detract from the paper's credibility.

**Justification For Why Not Lower Score:**

N/A

---

### Decision · Program_Chairs · 2024-01-16

Reject